# Wearable Technology for Monitoring Respiratory Rate and SpO_2_ of COVID-19 Patients: A Systematic Review

**DOI:** 10.3390/diagnostics12102563

**Published:** 2022-10-21

**Authors:** Shizuko Takahashi, Eisuke Nakazawa, Sakurako Ichinohe, Aru Akabayashi, Akira Akabayashi

**Affiliations:** 1Department of Biomedical Ethics, Faculty of Medicine, University of Tokyo, 7-3-1 Hongo, Bunkyo-ku, Tokyo 113-0033, Japan; 2Sanno Medical Center, 8-5-35 Akasaka, Minato-ku, Tokyo 107-0052, Japan; 3Department of Biochemistry and Biophysics, School of Medicine and Dentistry Center for RNA Biology, University of Rochester, 601 Elmwood Avenue, Rochester, NY 14642, USA; 4Division of Medical Ethics, School of Medicine, New York University, 227 East 30th Street, New York, NY 10016, USA

**Keywords:** wearable device, COVID-19, systematic review, monitoring, patients’ health status

## Abstract

With the significant numbers of sudden home deaths reported worldwide due to coronavirus disease 2019 (COVID-19), wearable technology has emerged as a method for surveilling this infection. This review explored the indicators of COVID-19 surveillance, such as vitals, respiratory condition, temperature, oxygen saturation (SpO_2_), and activity levels using wearable devices. Studies published between 31 December 2019, and 8 July 2022, were obtained from PubMed, and grey literature, reference lists, and key journals were also searched. All types of articles with the keywords “COVID-19”, “Diagnosis”, and “Wearable Devices” were screened. Four reviewers independently screened the articles against the eligibility criteria and extracted the data using a data charting form. A total of 56 articles were on monitoring, of which 28 included SpO_2_ as a parameter. Although wearable devices are effective in the continuous monitoring of COVID-19 patients, further research on actual patients is necessary to determine the efficiency and effectiveness of wearable technology before policymakers can mandate its use.

## 1. Introduction

Despite scientists trying to unravel the complexities of COVID-19, the end of the pandemic remains unforeseeable, as highly resistant variants continue to emerge. From the beginning of the pandemic, wearable devices have been used to conduct contact tracing, symptom screening, and routine testing, which are current COVID-19 public health surveillance methods [1]. Wearable technology refers to electronic devices worn on various parts of the body or built into clothing or accessories. Prior to the pandemic, wearable technology was used for the detection of other illnesses, such as neurological disorders and cardiovascular and respiratory diseases [2]. Although predominantly used for fitness tracking, the healthcare sector witnessed a proliferation of wearable technology owing to its medical applications and demands for systems to diagnose and track COVID-19 infections.

Approximately a year after the pandemic, wearable devices were shown to be effective in the early diagnosis of COVID-19 for symptomatic and asymptomatic patients [3]. A study by Mishera et al. showed that smartwatch data—heart rate (HR), number of daily steps, and sleep time—can detect pre-symptomatic cases of COVID-19 [4]. Additionally, they found that 81% of the participants could be identified as potentially COVID-19 positive, 4–7 days prior to the onset of the diagnosis. The researchers created an online detection algorithm that identified the early stages of infection through real-time heart rate monitoring.

Nevertheless, despite these efforts to pre-diagnose COVID-19, the rapid spread of the disease led to a shortage of hospital beds, medical supplies, and healthcare workers. Managing a large number of patients using intensive care units has been challenging. Therefore, continuous monitoring of patients with physiological parameters is required. Numerous patients are self-quarantined at home or in self-isolation. Although the vast majority of patients are able to manage COVID-19 symptoms at home, sudden drastic changes in symptoms, some even leading to death, remain a challenge for disease control. Unexpected home deaths from COVID-19 persist and remain a mystery since no specific characteristics could be identified for these patients. A 2021 excess mortality modeling analysis estimated an additional 24% of unrecognized COVID-19-attributable deaths [5].

Since 2021, popular smartwatches produced by Apple, Samsung, Huawei, Garmin, Fitbit, and others have integrated the function of measuring oxygen saturation (SpO_2_) [S31] (selected references in Appendix A are numbered with S). Health parameters for COVID-19 can be monitored through heart rate variability (HRV), SpO_2_, respiration rate, temperature, and lung capacity [S25]. The oxygen saturation and respiration rate have received special attention since the WHO guideline suggests that patients with an SpO_2_ greater than 94% can be cared for at home, while those with levels below 93% should receive medical attention, as it could indicate a moderate–severe case of COVID-19 [6]. Hypoxia in patients with the disease has been shown to be a strong predictor of mortality [7]. Silent hypoxia has been recognized as evidence of impending deterioration. Therefore, it is crucial to develop an effective monitoring system for COVID-19 patients using wearable devices that can detect instant changes in symptoms to prevent sudden deaths occurring from unexpected changes in the patient’s health status. Many countries have prematurely employed autonomous self-measurement of SpO_2_ by distributing pulse oximeters to those confirmed to have COVID-19, without having undertaken a proper randomized controlled trial, the gold standard of evidence.

A recent randomized controlled trial by Lee et al., using outcomes from more than 2000 patients monitored at home, revealed that using a pulse oximeter to measure oxygen levels was no better than regularly asking patients with COVID-19 if they are short of breath [S34]. However, their study does not dismiss SpO_2_ as a parameter entirely, since patients in the study were asked to submit readings intermittently, and SpO_2_ was not measured continuously. Additionally, considering the large number of symptomatic survivors of COVID-19, identifying tools and instruments that can be deployed remotely, conveniently, and with the patient at home can help monitor patients suffering from a slow or incomplete recovery or those with worsening symptoms over time [S1]. Simultaneously, possible secondary infections within family members and caregivers may go undiagnosed in some countries due to a lack of testing facilities. Therefore, in this study, we examined the utility of wearable devices to continuously monitor SpO_2_ and respiratory rate in COVID-19 patients after diagnosis.

## 2. Materials and Methods

This systematic review used PubMed as the database, with the keywords “COVID-19”, “Diagnosis”, and “Wearable Devices”. The search was limited to studies published between 31 December 2019, and 8 July 2022, as the pandemic occurred during this period. After the search, 275 articles were identified and filtered, as shown in Figure 1. All 275 articles were read by the authors, filtered, and categorized. Of these, 42 were first removed since they focused primarily on the devices themselves. Thereafter, 142 were removed since they did not pertain to COVID-19. Then, we filtered out 91 articles based on the title and the abstract. In the majority of these 91 articles, blood pressure, SpO_2_, temperature, and activity were measured, so we classified them according to how these items were used. These were categorized into “diagnostic” use (33 articles), “monitoring” use (56 articles), and “treatment” use (2 articles). “Diagnostic” use is when the above items are intended to be used for diagnosis. For example, an increase in blood pressure or body temperature may indicate COVID-19, or the device may be used to avoid the risk of COVID-19 infection, making online calls to the hospitals more efficient. Moreover, they can be used to remotely detect changes in the blood pressure or SpO_2_ of an existing patient, i.e., the devices are used to find infected patients. The term “monitoring” is used when the above items are intended, for example, to detect the drop in SpO_2_ of a COVID-19 patient. In this case, the device is used to obtain information to change the therapeutic process (e.g., medications, ventilators, etc.). The term “treatment” is used when the device is used for therapeutic purposes. Different to “monitoring”, “treatment” refers to when the device is used to track the progress of a disease (e.g., to measure the motor function of a patient with Parkinson’s disease), and then to alert the patient to walk or exercise more as part of the treatment intervention. After the “treatment use” (alert), the progress will be monitored by “monitoring” use.

The four parameters (respiratory, circulatory, activity, and temperature) used in 91 articles are shown in Figure 2. Respiratory (n = 30): when targeting respiratory illnesses including COVID-19, or patients suffering from COVID-19; circulatory (n = 36): to measure the heartbeat of patients with heart disease, hypertension, etc.; activity (n = 25): to measure whether or not the patient’s movement is decreasing due to COVID-19 infection or to track patients with neurological diseases such as Parkinson’s disease; temperature (n = 23): many of the devices measure the body temperature of COVID-19 infected patients, and that of other respiratory, cardiac, and neurological diseases.

We focused on “respiratory” articles since they contain the most relevant information about wearable devices for measuring SpO_2_, hand-searching four articles on pulse oximeters using Google Scholar. There are papers that have more than one parameter in a single paper, resulting in an overlap of 118 papers, as shown in Figure 2. The actual number of papers in Figure 2 is 95 (91 plus 4). We finally focused on the 34 articles on “respiratory indicators” (Appendix A).

## 3. Results and Discussion

The 34 articles on “monitoring use” were categorized into four subject areas: on the wearable sensors themselves; on patient processing framework data derived from the devices; on the use of wearable devices on actual COVID-19 patients; and on operational concerns.

### 3.1. Wearable Sensors

With the pandemic, the demand to diagnose and monitor COVID-19 infections non-invasively and continuously has intensified interest in consumer-grade wearables. We focused on respiratory wearables, since lack of oxygen, that is, SpO_2_ (<95%), and increased respiratory rate (≥30 bpm) are the most common indications of health deterioration and could result in brain damage, heart failure, or sudden death, and acute respiratory distress syndrome (ARDS) [S6, S11]. These devices were used with the underlying belief that having continuous biometric data can help detect subtle changes, which could, in turn, be used for the early detection of COVID-19, prior to symptom onset apparent to the individual. Furthermore, they can also be used to monitor impending deterioration after onset of infection.

Out of the 39 articles on respiratory indicators, six were reviews on available wearable sensors [S6, S9, S11, S25, S27, S28]. A summary of wearable technology for monitoring patients with COVID-19 was published by Ding et al. [S11]. A smartwatch, ring, wrist-worn band, earbud, and flexible skin-like e-tattoos were reviewed for measuring SpO_2_, and a chest/abdominal strap, vest, facial mask (with a humidity sensor), and flexible patch were reviewed for measurement of respiratory rate (RR) [S9, S11]. Tayal designed their own device to measure SpO_2_, respiration rate, pulse rate, and body temperature [S12]. Jiang et al. designed a prototype of a fully integrated chest strap that also measured ECG, HRV, pulse pressure wave, blood pressure, cough frequency, and lung volume [S25]. However, this device seemed quite elaborate for practical use. Channa et al. reviewed the price of wearable devices, such as a wristband with a finger clip, and a finger clip only, which cost USD 112 and USD 299, respectively [S6]. Popular commercial devices include the Oura ring, Fitbit, Apple Watch, Garmin, and WHOOP wrist-worn strap [S28]. The features that differentiate one device from another are long-term monitoring technology, battery life, device reusability, multi-modal symptom detection, and cost [S6].

Channa et al. also showed multiple applications of wearable devices during the pandemic, including preventing worsening of the disease, quarantine management, effective contact tracing, social or business interactions, smart learning/education, diagnosis of COVID-19, and stress management [S6]. Various researchers have proposed that physiological information obtained from integrated wearable devices, which have the utility of artificial intelligence (AI) and sensor fusion techniques, can have promising results, and are potentially ideal for the long-term monitoring of COVID-19 patients and other chronic diseases [8]. These are advantageous since the sensors are minimized and integrated into the network connectivity and predictive analytics to capture, transmit, and analyze biometric information automatically [9]. With its ability to continuously generate real-time measurements, wearable technology requires minimal involvement from users and healthcare professionals, thereby minimizing transmission of the virus [10]. The increased use of wearable devices can be attributed to their multi-functional and versatile application [11].

### 3.2. COVID-19 Patient Monitoring Framework (Obtained from the Device)

The majority of articles mentioned contact tracing as a part of the monitoring framework, using GPS information and data obtained from the wearable device. Different types of Internet of Things (IoT) architectures have been proposed and investigated. The three-layered approach was observed the most and was mentioned in five articles [S10, S12, S22, S26, S29].

#### 3.2.1. First Layer

The first is the sensors layer, or the wearable IoT layer, which is located on various parts of the body. These come in two types: GPS sensor-based data and health-related information, such as temperature, HR, SpO_2_, and cough count systems. The inclusion of data from various sensors into the algorithmic decision-making process could reduce the effects of environmental, behavioral, and other external factors (such as diet, travel, alcohol, stress, drug intake, and other health conditions) on diagnostic outcomes [3]. The function of this layer is that it can sense the abnormal symptoms of the patients and collect GPS data to help locate the patient and monitor them in a more timely manner [S22]. Battery life and security of the data transmitted are of great importance [S26].

#### 3.2.2. Second Layer

This layer, the cloud layer, is responsible for receiving patients’ real-time data from the microcontroller stored in the cloud by establishing basic security measurements [S10, S22, S26]. Cloud flare is often used to develop the security of the data via Internet connections, such as Wi-Fi or ground-penetrating radar (GPR). The successive establishment of security factors maintains the healthcare application’s data reliability, scalability, and storage [S10]. All the details of patient symptoms, emergency contact, and location data are stored in Cloud flare’s global network [S22]. These data are transferred to authorized users via the authorized programming interface endpoints [S10, S22]. Once COVID-19 affects a patient, they are alerted by SMS and email regarding the proper course of action.

#### 3.2.3. Third Layer

This layer, the network layer, receives real-time data from the cloud system by maintaining credibility and data ownership [S10, S22]. This layer connects the data to the outside world through data processing and data application [S26]. In this way, it can also be used to analyze the information and alert the authorities from time to time and call in an emergency to the patient’s response team [S22]. The biggest concern for this layer is the security of the data. As it manages communications between personal devices and other services, such as hospitals, it is more vulnerable to exploitation. Data protection is a challenge, and new approaches are being developed.

### 3.3. Data from Actual Patients

There is not sufficient evidence to suggest that solely measuring SpO_2_ continuously through a wearable device is effective in terms of COVID-19 patient outcomes. While observational studies using various devices and various parameters, including respiration rate (RR) and SpO_2_, showed that changes were observed, they were not shown to be indicators of deterioration. Nevertheless, monitoring RR, SpO_2_, HR, and temperature at home using a wearable device was reported to have shortened the time taken to visit a hospital—90% of the hospitalized patients indicated that they would have delayed hospitalization further if they had not been a part of the study [S13]. Re-admission is not only due to the data from the devices but also to the symptoms. A study by Patel et al. showed that re-admission to the hospital (4%) or to the emergency department (10%) was for chest pain, shortness of breath, and persistent fever [S3]. A study by Wurzer et al. showed that 13% of patients who were referred to the hospital by the TeleCOVID team not only had low SpO_2_ values of less than 88.0% (as compared to the non-hospitalized group with 96.5%), but were also more symptomatic and showed dyspnea and fever more frequently.

The majority of participants felt safe [S13] about having joined the study with wearable devices. Similarly, a study by Bircher et al. showed that in a virtual maternity ward, monitoring patients remotely using finger pulse oximetry intermittently, or using a wearable device continuously, offered COVID-19-positive patients a reassurance of safety. Whether these patients truly benefited from the data obtained from the wearable device remains to be shown.The characteristics of the studies included are presented in Table 1.

Alboskmaty et al., recently reviewed 13 articles on the effectiveness of pulse oximetry in monitoring patients with COVID-19 and found a reduction in the number of hospital readmissions and visits [S32]. Their conclusion was the same as our results, in that the use of pulse oximetry can potentially save hospital resources for patients who might benefit most from care escalation. However, there was no evidence regarding the effect of remote patient monitoring with pulse oximetry on health outcomes, as compared with other monitoring models.

### 3.4. Operational Concerns

The accuracy of pulse oximetry can be influenced by multiple factors, including skin perfusion and pigmentation, and the anatomical variability of the wrist, which was quoted as challenges. Even under ideal conditions, Spaccartotella et al., could not measure oxygen levels with a smartwatch. Permanent or temporary changes to the skin, such as tattoos, are another factor that may affect measurements [S5]. The ink used in some tattoos, and their design and saturation, can block light from the sensor, thereby preventing the O_2_ levels app from taking accurate measurements. Furthermore, racial bias in pulse oximetry measurements has been raised; Sojoding et al. examined two large cohorts and found that Black patients had nearly three times the frequency of occult hypoxemia not being detected by pulse oximetry as White patients [S33].

They suggest that reliance on pulse oximetry to triage patients and adjust supplemental oxygen levels may place Black patients at increased risk of hypoxemia. They urge consideration of variation in risk according to race and a correction of racial bias in pulse oximetry.

The most popular wearables, such as Fitbit sensors and the Oura Ring, have not been cleared by the U.S. Food and Drug Administration for remote monitoring. They do not offer pulse oximetry, body temperature, or high-fidelity measurements of respiratory rate [S27]. An application to install these measurements is required. Commercial wearables are not intended to replace clinical diagnostic tools, but rather are to be used for recreational purposes. Therefore, although these devices have the potential to detect, predict, and monitor patients with COVID-19, the specifications are insufficient to replace what is clinically available. Furthermore, as Ates et al., indicated, we were unable to find a report of a device that is able to differentiate COVID-19 from other viral infections [3]. Although they indicated that they could not find a report, we think that if a health provider gives people rapid test kits with wearable devices, when a fever, etc. is detected, the device will alert the person immediately to examine using the rapid test kit.

There are also problems with the practicalities of wearing the device itself. In Werzer’s study, only two-thirds of those enrolled reported positively or somewhat positive about the comfort of wearing an in-ear monitoring device [S13]. There is also an age restriction for wearing certain devices. For example, children with smaller wrists were unable to use wrist band type devices. The elderly also have reservations about wearing these devices and are predisposed to sample bias, as older people and low-income populations do not commonly own wearable devices. These issues could be resolved with lowered costs and access to in-person information for these devices.

Goergen et al. reviewed the advantages and disadvantages of wearable devices and biometric data, reporting limitations related to data storage to be a disadvantage of micro-controller units, smartphones, and cloud processing. Additionally, access to the internet and batteries are of concern while using these devices. Therefore, the developing world faces more challenges concerning Biomet interoperability and real-time processing during COVID-19 outbreaks [S29].

## 4. Conclusions

Wearable devices can potentially save hospital resources for patients who might benefit most from care escalation, although evidence on the effects of remote patient monitoring with pulse oximetry and respiratory rate on health outcomes remains to be found. Randomized controlled trials may be difficult to perform since people have become so used to managing COVID-19 infections using pulse oximeters. Wearable devices, including those focused on other parameters and adjusted to a greater variety of populations (e.g., racial diversity and younger age), are necessary.

## Figures and Tables

**Figure 1 diagnostics-12-02563-f001:**
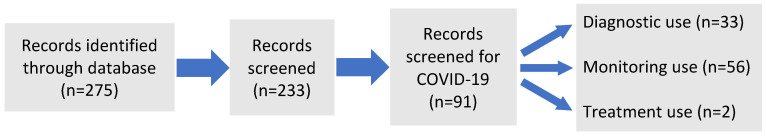
Diagram of systematic selection by use.

**Figure 2 diagnostics-12-02563-f002:**
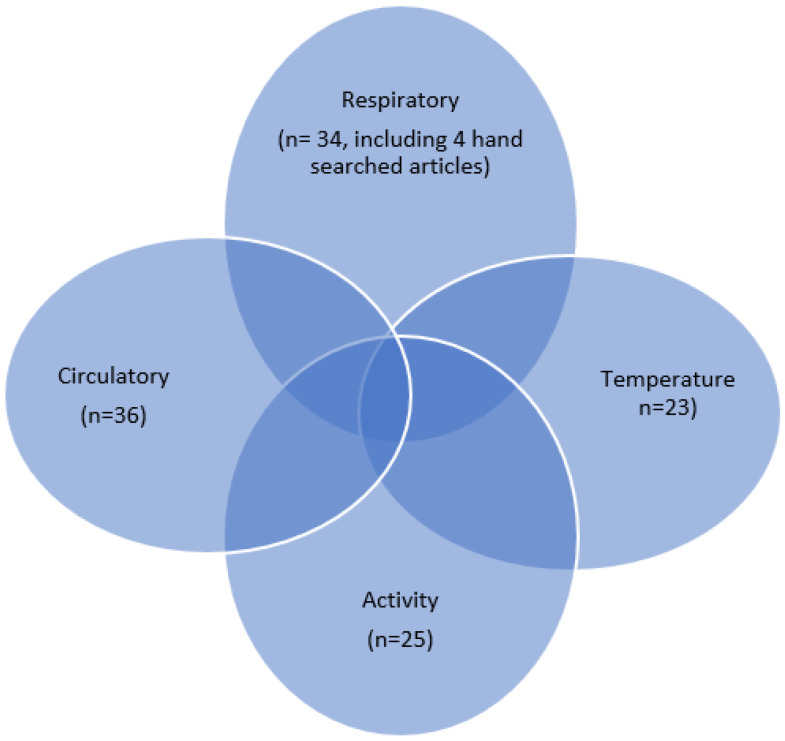
Diagram of systematic selection by parameters.

**Table 1 diagnostics-12-02563-t001:** List of articles on wearable devices using actual patients.

No. in Appendix A	Country	Publish Date	Study Period	Aims and Objectives	Sample Size
S20 Gielen et al. (2021)	USA	October 2021	April–June 2020	To observe marked changes in biometric measurements (HR, RR, SpO_2_, arterial stiffness) around the dates of infection	933 subjects with 2 detected with COVID-19 infection
S30 Laveric et al. (2022)	Romania	January 2022	Not specified	To test the performance of an IoT device, which can monitor health parameters such as HR, SpO_2_, body temperature, and current location	2 persons under quarantine
S24 Un et al. (2021)	China	February 2021	19 March 2020–11 April 2020	Observational study of mild COVID-19 patients with wearable biosensors (HR, HRV, RR, SpO_2_, BPW, temperature, actigraphy) and machine learning-based remote monitoring	34 COVID-19 positive patients
S23 Santos et al. (2021)	UK	September 2021	March–August 2020	Adaption of a wearable-based ambulatory monitoring systems for real-time remote monitoring of the vital signs of COVID-19 patients cared for at an isolation ward	59 COVID-19 positive patients
S8 Hussain et al. (2022)	Oman	January 2022	Not specified	The article prediction analysis is carried out with the dataset downloaded from the application peripheral interface (API), designed explicitly for COVID-19 quarantined patients	1085 patients (490 COVID-19 infected and 595 non-infected cases)
S21 Mekhael et al. (2022)	USA	March 2022	~September 2021	Assessment of the long-term effects of COVID-19 through sleep patterns from continuous signals collected via wearable wristbands	122 patients with COVID19 and 588 controls (n = 710).
S3 Patel et al. (2022)	USA	February 2022	April–June 2020	To describe a pilot program to evaluate the impact of remote patient monitoring in post-discharge monitoring of COVID-19 patients	80 high-risk COID-19 patients discharged from the hospital
S13 Wurzer et al. (2021)	Germany	September 2021	None Specified	To establish a telemonitoring system for COVID-19 positive high-risk patients in domestic isolation	153 patients, older than 60 years of age and with a pre-existing condition
S2 Bircher et al. (2022)	UK	July 2022	October 2021–February 2022	To determine appropriate virtual care and telehealth systems to reduce barriers to care and improve maternity outcomes	228 COVID-19 maternity patients admitted to the virtual maternity ward
S34 Lee et al. (2022)	USA	May 2022	29 November 2020–5 February 2021	To determine the effectiveness of home pulse oximetry monitoring on COVID-19 patients	A total of 1041 patients (606 COVID-19 positive) as control, and 1056 patients (611 COVID-19 positive) were in the pulse oximetry group

## Data Availability

Not applicable.

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
