# Peer review of "Wearable Technology for Monitoring Respiratory Rate and SpO2 of COVID-19 Patients: A Systematic Review"

_diagnostics, 2022, doi:10.3390/diagnostics12102563_

Round 1

Reviewer 1 Report

The authors present a systemic review on the use of the respiratory rate and the spontaneous oxygen saturation to help in the management of Covid 19.  The authors found that these monitoring method reduce the hospital resources utilized by patient who need escalation in health intervention but were unable to determine any effect on health outco

Author Response

Once again, thank you for reading our manuscript in detail.

Reviewer 2 Report

Authors discuss an important aspect of medicine that is currently in development.  The authors clearly state the lack of consistency and accuracy in the data obtained from wearable devices. However, they do bring to the attention of the readers of the ongoing efforts to improve the technology where it can become assistive technology to clinical diagnosis. As the testing and lack of reporting of COVID-19 positive cases wanes world-wide, the wearable tracking technology could be immensely helpful to alert patients and medical community on new SARS-CoV-2 variants and other yet-to-be-identified pathogens.  Overall, authors capture the recent developments in the field.  I have following comments for the authors:

Lines 85 - 102: Describe in detail how the search was performed. Were the articles categorized by search terms or were they read by the authors? I am confused about the “Hand search” -term in figure 2 and on line 96.  Explain what criteria was used to remove 42 articles.  In text authors mention “not applicable” but this does not explain why. Similarly, please, explain how the articles were binned to “diagnostic”, “monitoring”, and “treatment use”. These three categories are not obvious to a reader but rather appear overlapping. I presume that 56 articles (line 93) are referring to overlapping categories n=36, n=39, n=23, and n=25.  I would suggest to the authors to use a Venn diagram in Figure 2 instead of a flowchart. 

Line 160: Energy – unclear meaning

Line 173: Web fronted layer – use another term

Line 179: Securitization – use another term

Line 279-280: This is only one reference and not two references

I would like to see a discussion how wearable devices can distinguish COVID-19 from other acute or chronic respiratory illnesses.

Author Response

Reply to Rev 2.  Comments and Suggestions for Authors

Authors discuss an important aspect of medicine that is currently in development.  The authors clearly state the lack of consistency and accuracy in the data obtained from wearable devices. However, they do bring to the attention of the readers of the ongoing efforts to improve the technology where it can become assistive technology to clinical diagnosis. As the testing and lack of reporting of COVID-19 positive cases wanes world-wide, the wearable tracking technology could be immensely helpful to alert patients and medical community on new SARS-CoV-2 variants and other yet-to-be-identified pathogens.  Overall, authors capture the recent developments in the field.  I have following comments for the authors:

Lines 85 - 102: Describe in detail how the search was performed.

# We have significantly revised the text and corrected the number of Figure 1.

This systematic review used PubMed as the database, with the keywords “COVID-19,” “Diagnosis,” and “Wearable Devices.” The search was limited to studies published between December 31, 2019, and July 8, 2022, as the pandemic occurred during this period. After the search, 275 articles were identified and filtered, as shown in Figure 1. All 275 articles were read by the authors, filtered, and categorized. Of these, 42 were first removed since they focused primarily on the devices themselves. Thereafter, 142 were removed since they did not pertain to COVID-19. Then, we filtered out 91 articles based on the title and the abstract. In the majority of these 91 articles, blood pressure, SpO2, temperature, and activity were measured, so we classified them according to how these items were used. These were categorized into “diagnostic” use (33 articles), “monitoring” use (56 articles), and “treatment” use (2 articles).
“Diagnostic” use is when the above items are intended to be used for diagnosis. For example, an increase in blood pressure or body temperature may indicate COVID-19, or the device may be used to avoid the risk of COVID-19 infection, making online calls to the hospitals more efficient. Moreover, they may be used to remotely detect changes in the blood pressure or SpO2 of an existing patient, i.e., the devices are used to find infected patients. The term “monitoring” is used when the above items are intended, for example, to detect the drop in SpO2 of a COVID-19 patient. In this case, the device is used to obtain information to change the therapeutic process (e.g., medications, ventilators, etc.). The term “treatment” is used when the device is used for therapeutic purposes. Different to “monitoring”, “treatment” refers to when the device is used to track the progress of a disease (e.g., to measure the motor function of a Parkinson's disease patient), and then to alert the patient to walk or exercise more as part of the treatment intervention. After the “treatment use” (alert), the progress will be monitored by “monitoring use.”

The four parameters (respiratory, circulatory, activity, and temperature) used in 91 articles are shown in Figure 2. Respiratory (n=30): when targeting respiratory illnesses including COVID-19, or patients suffering from COVID-19; circulatory (n=36): to measure the heartbeat of patients with heart disease, hypertension, etc.; activity (n=25); to measure whether or not the patient's movement is decreasing due to COVID-19 infection or to track patients with neurological diseases such as Parkinson's disease; temperature (n=23): many of the devices measure the body temperature of COVID-19 infected patients, and that of other respiratory, cardiac, and neurological diseases.

We focused on “respiratory” articles since these contain the most relevant information about wearable devices for measuring SpO2, hand-searching four articles on pulse oximeters using Google Scholar. There are papers that have more than one parameter in a single paper, resulting in an overlap of 118 papers, as shown in in Figure 2. The actual number of papers in Figure 2 is 95 (91 plus 4). We finally focused on the 34 articles on “respiratory indicators” (Appendix 1).

Were the articles categorized by search terms or were they read by the authors?

#Read by authors

I am confused about the “Hand search” -term in figure 2 and on line 96. 

 Explain what criteria was used to remove 42 articles.

In text authors mention “not applicable” but this does not explain why.

Similarly, please, explain how the articles were binned to “diagnostic”, “monitoring”, and “treatment use”. These three categories are not obvious to a reader but rather appear overlapping. I presume that 56 articles (line 93) are referring to overlapping categories n=36, n=39, n=23, and n=25. 

# We hope above revision has answered all of your questions.

  I would suggest to the authors to use a Venn diagram in Figure 2 instead of a flowchart. 

#Yes, we have added Venn diagram

Line 160: Energy – unclear meaning

#Battery-life.

Line 173: Web fronted layer – use another term

#Network layer.

Line 179: Securitization – use another term

#protecting data, or data protection.

Line 279-280: This is only one reference and not two references

#Yes, you are right. [7] is a single reference

Xie, J.; Covassin, N.; Fan, Z. et al. Association Between Hypoxemia and Mortality in Patients With COVID-19. Mayo Clin Proc 2020;95(6):1138-1147. doi: 10.1016/j.mayocp.2020.04.006.

I would like to see a discussion how wearable devices can distinguish COVID-19 from other acute or chronic respiratory illnesses.

#Thank you. We have added Ates et al.‘s discussion [3].

Furthermore, as Ates et al. indicated, we were unable to find a report of a device that is able to differentiate COVID-19 from other viral infections [3]. Although they indicated that they could not find a report, we think that if a health provider gives people rapid test kits with wearable devices, when a fever, etc. is detected, the device will alert the person immediately to examine using the rapid test kit.

Once again, thank you for reading our manuscript in detail.